# A Comparison of Central Screw versus Post for Glenoid Baseplate Fixation in Reverse Shoulder Arthroplasty Using a Lateralized Glenoid Design

**DOI:** 10.3390/jcm11133763

**Published:** 2022-06-29

**Authors:** Michael J. Bercik, Brian C. Werner, Benjamin W. Sears, Reuben Gobezie, Evan Lederman, Patrick J. Denard

**Affiliations:** 1Lancaster Orthopedic Group, Lancaster, PA 17601, USA; michaelbercik@gmail.com; 2Department of Orthopaedic Surgery, University of Virginia, Charlottesville, VA 22903, USA; bcw4x@virginia.edu; 3Western Orthopaedics, Denver, CO 80218, USA; bwsears@gmail.com; 4The Cleveland Shoulder Institute, Beachwood, OH 44194, USA; clevelandshoulder@gmail.com; 5Banner Health, Phoenix, AZ 85012, USA; elederman1@icloud.com; 6Oregon Shoulder Institute, Medford, OR 97504, USA

**Keywords:** reverse shoulder arthroplasty, baseplate fixation, longevity, clinical outcomes

## Abstract

The purpose of this study was to compare the short-term clinical and radiographic outcomes of a lateralized glenoid construct with either a central screw or post. **Methods:** A multicenter retrospective study was conducted of reverse shoulder arthroplasties (RSAs) with minimum 2-year clinical followup. All RSAs implanted had a 135° neck shaft angle (NSA) and a modular circular baseplate. The patients were divided into two cohorts based on the type of central fixation for their glenoid baseplates (central post (CP) vs. central screw (CS)). The clinical outcomes, rates of revisions, and available radiographs were evaluated. **Results:** In total, 212 patients met the study criteria. Postoperatively, both groups improved over their preoperative baseline. There were no significant differences between the cohorts in any PROs at 2 years postoperatively. No findings of gross loosening were identified in either cohort. Implant survival was 98.6% at 2 years. **Conclusions:** When using a lateralized glenoid implant with a 135° NSA inlay humeral component, both central post and central screw baseplate fixation provide good clinical outcomes, survivorship, and improvements in ROM at 2 years. There is no difference in loosening or revision rates between the types of baseplate fixation at a minimum of 2 years postoperatively.

## 1. Introduction

Reverse total shoulder arthroplasty (RSA) provides excellent pain relief and improved function for patients with a wide and expanding variety of shoulder pathologies. Given its success, RSA has increased in use and now is more frequently utilized than anatomic shoulder replacement [1]. Given the increasing incidence of implantation, as well as increased use in the younger, more active patient population, long-term survivorship remains a critical goal.

RSA has demonstrated survivorship rates of 76–97% at 8 to 10 years [2,3,4,5]. These studies represent a heterogeneous study population, however, with varying neck shaft angles (NSA) and means of glenoid baseplate central fixation (i.e., central post versus central screw) amongst other factors that vary between the populations. Comparative studies of central post versus screw constructs are lacking. While baseplate loosening is uncommon regardless of fixation with a medialized construct, lateralization increases stress on the baseplate, which may increase the risk of failure [6]. At the same time, it has increasingly been shown that glenoid lateralization improves outcomes compared to a medialized construct [7,8]. It is therefore important to compare central fixation—i.e., a central post versus screw—in the setting of lateralized glenoid constructs.

The purpose of this study was to compare the short-term clinical and radiographic outcomes of a lateralized glenoid construct with either a central screw or post. We hypothesized that there would be no clinical or radiographic differences between a central post or screw construct at the 2-year followup.

## 2. Materials and Methods

### 2.1. Database and Study Patients

Institutional review board approval was obtained prior to beginning the study. Data were retrospectively reviewed from a multicenter prospectively maintained database of shoulder arthroplasties. This database was retrospectively queried for RSAs performed between August 2018 and August 2019. Demographic and patient information were collected. The inclusion criteria were as follows: (1) primary RSA; (2) baseline range of motion measurements (ROMs) and patient-reported outcomes (PROs); (3) diagnoses of rotator cuff arthropathy or glenohumeral joint arthritis; and (4) minimum clinical and radiographic followup of two years. Exclusion criteria when studying for survivorship included (1) lack of 2-year data; (2) diagnoses other than rotator cuff arthropathy or glenohumeral joint arthritis; and (3) concomitant glenoid bone grafting. Exclusion criteria when studying for functional analysis included (1) revision RSA; (2) diagnoses other than rotator cuff arthropathy or glenohumeral joint arthritis; and (3) concomitant glenoid bone grafting. After selection of the patient population, patients were further grouped based on their central fixation into either a central screw (CS) or central post (CP) cohort. The management of the subscapularis tendon (i.e., peel, tenotomy, or lesser tuberosity osteotomy) was also recorded.

### 2.2. Surgical Technique

Surgeries were performed by 11 fellowship-trained experienced surgeons who all perform more than 25 RSAs a year. A deltopectoral incision was used in all cases. The choice of subscapularis management was made by each surgeon based on preference. The humerus was cut at 135° and a press-fit short or standard length inlay humeral component was implanted in all cases (Apex Revers or Univers Revers; Arthrex, Inc., Naples, FL, USA). On the glenoid side, a modular circular baseplate was used in all cases (Modular Glenoid System; Arthrex, Inc.). This allows for the use of either a central screw or central post of variable length (15–35 mm) (Figure 1). Additionally, the baseplate comes with offset option of 0, 2, or 4 mm of lateralization that may be combined with a glenosphere of 0 or 4 mm of lateralization. Glenosphere diameter options include 33, 36, 39, or 42 mm. Central component selection (post vs. screw), central component length, glenosphere diameter, and total lateralization were all based on surgeon preference with the goal of restoring appropriate center of rotator and muscle tension. Software planning was also utilized based on surgeon preference.

### 2.3. Clinical Outcomes

The clinical outcome data included both patient surveys and objective clinical measurements. Surveys administered were: (1) American Shoulder and Elbow Surgeons (ASES) questionnaire, (2) Visual Analog Scale (VAS) for pain, (3) Single Assessment Numeric Evaluation (SANE), (4) Constant-Murley score, (5) Western Ontario Osteoarthritis Shoulder (WOOS) index, and (6) VR-12 Mental outcome measure. Objective clinical measurements were the following range of motion (ROM) measurements obtained at preoperative and postoperative clinic visits: active forward flexion (AFF), external rotation (ER) at the side, ER at 90 degrees, internal rotation (IR) measured as the highest spinal level achieved, and IR at 90 degrees of shoulder abduction. Rate of revision at 2-year followup was also calculated, as was the reason for revision.

### 2.4. Radiographic Evaluation

All available 2-year postoperative images were evaluated by two fellowship-trained surgeons for loosening using the criteria described by Melis, et al. [9]. Specifically, any radiolucent lines around the glenoid screws, around the peg, or below the baseplate were classified according to their width (<2 mm or ≥2 mm). Loosening was considered to be present if the glenoid component had migrated, as demonstrated by shift, tilt, or subsidence, or if complete radiolucency ≥ 2 mm was present in each zone.

The β-angle of the glenoid baseplates relative to the floor of the supraspinatus fossa was also measured on an anteroposterior radiograph for each patient [10].

### 2.5. Statistical Analysis

Comparisons of continuous variables such as age, BMI, baseline and 2 year PROs and ROM were performed using Student’s *t*-tests. Comparisons of the remaining categorical variables were performed using chi-squared tests. For all comparisons, *p* < 0.05 was considered statistically significant. All analyses were performed in SPSS version 28 (IBM, Armonk, NY, USA).

## 3. Results

### 3.1. Patient Demographics

In total, 212 patients met the study criteria (Table 1). The mean age of the entire study population was 68.2 ± 8.3 years. Male patients compromised 54.7%, and the dominant arm was involved in 60.8% of patients. A total of 125 patients underwent fixation with a central screw, and 87 patients underwent fixation with a central post. There were several baseline differences noted. In the CP cohort, males compromised 63.2% of patients whereas in the CS cohort they comprised 48.8% (*p* = 0.038). A 33 mm glenosphere was utilized in 22.4% of cases utilizing a central screw and in 10.3% of cases involving a central post (*p* = 0.023). Lastly, a larger percentage of patients in the CP group underwent tenotomy (89.7% versus 62.4%; *p* < 0.001).

Glenoid lateralization was not significantly different between the two groups. The majority of cases in both groups involved either 6 mm (47.2% CS vs. 40.2% CP) or 8 mm (28% CS vs. 35.6% CP) of lateralization. A lateralization of 4 mm was noted in 21.6% of CS cases and 23% of CP cases. A lateralization of 2 mm was used in 3.2% of CS cases and 1.1% of CP cases. The most frequently utilized screw length was 25 mm (range: 15–35 mm), and the most frequently utilized post length was 20 mm (range: 15–30 mm) (*p* ≤ 0.001). The remaining demographic information revealed no other significant baseline differences (Table 1).

### 3.2. Clinical Outcomes

Three revisions were excluded from the two year outcome analysis, leaving 209 patients available for review. Preoperatively, the CP cohort had a higher average SANE score (37.2 vs. 28.5; *p* = 0.004), and the CS cohort had greater active IR at 90 degrees (27 vs. 18; *p* = 0.006). Otherwise, there were no significant clinical differences in terms of baseline PRO or ROM between the two groups (*p* > 0.05; Table 2).

Postoperatively, both groups demonstrated significant improvement over their preoperative baselines. There were no significant differences in the PROs reported for either cohort at 2 years postoperatively (Table 3). When comparing ROM at 2 years, no significant differences were noted except increased active FF (142 vs. 133; *p* = 0.003) and increased IR in the CP group (2 vs. 0; *p* < 0.001). A comparison of change from preoperative to postoperative (Table 4) revealed no differences in PROs but significant differences in increased active ER at the side in favor of the CS group (24 vs. 15; *p* = 0.002) and increased active IR in the CP group (2 vs. 0; *p* < 0.001).

### 3.3. Radiographic Outcomes

Overall, 99 patients of the 209 patients had full postoperative X-ray available at the 2-year followup. Of these, 58 had a central screw, and 41 had a central post. No findings of gross loosening were identified in either cohort.

The β-angles in the two groups were +3° (range: 70–109°) (indicating superior tilt relative to the floor of the supraspinatus fossa) and −1° (range: 63–103°) (indicating inferior tilt relative to the floor of the supraspinatus fossa) in the CS and CP group, respectively (*p* = 0.022).

### 3.4. Complications and Implant Survival

Implant survival was 98.6% at 2-year minimum followup. One revision was performed in the CS group vs. two in the CP group (0.8% vs. 2.3%; *p* = 0.218). The revision for the patient with a central screw involved a liner exchange for a dislocation that occurred in the immediate post-operative period. One revision in the CP group was performed for glenoid loosening after the patient had sustained a fall and subsequent posttraumatic baseplate failure. This was revised with a structural glenoid bone graft. The other revision was performed for a periprosthetic fracture. There were no other implant complications in either group.

## 4. Discussion

The primary findings of the current study were that revision remained low with either a central post or screw construct, and the functional outcomes were similar between designs at early followup. Although initially reserved for elderly patients with rotator cuff arthropathy, the indications for RSA have expanded to numerous etiologies and younger patients. In an attempt to reduce scapular notching frequently seen with the Grammont design and to theoretically improve range of motion, some have advocated for a lateralized glenoid implant with a more anatomic NSA of 135 degrees. In an in vitro study, Gutierrez et al. found that a lateralized center of rotation offset had the largest effect on range of motion, whereas the neck shaft angle was most important to scapular notching rates [7]. In a demonstration of the clinical benefit of lateralization, Werner et al. demonstrated glenoid lateralization of 6–8 mm was associated with improved active IR at one year compared to patients with less glenoid lateralization [8]. No significant differences were noted in active forward flexion, external rotation, or PROs in their study.

There is little evidence to date that evaluates clinical differences based on form of central glenoid baseplate fixation. This was the impetus for the present study comparing clinical outcomes for patients in whom the glenoid baseplate was fixed with a central screw versus a central post for RSA. The results showed similar short-term improvements in both PROs and measured ROM in which both the CS and CP groups demonstrated significant improvement. Statistically significant differences were noted in the 2-year active IR that favored the CP group, but this may not have reached clinical significance. As neither the post nor the screw affects the lateralization or angle of the reverse implant, it was not anticipated that there would be significant clinical differences between the two groups.

While lateralization imparts clinical benefits that are becoming better understood, this increased lateralization also comes with the concern for increased destabilizing torque and subsequent loosening at the glenoid baseplate interface [11]. Despite these theoretical concerns, however, several studies have demonstrated excellent survivorship of lateralized implants. For example, Cuff et al. demonstrated 90.7% 10-year survivorship when evaluating a lateralized RSA with a 135 degree NSA and a baseplate with central screw fixation [3]. Finite element analysis has suggested the central screw constructs reduce micromotion compared to central post constructs [12]. Clinical studies that compare the stability of a lateralized implant with central post or screw fixation are lacking. In our study, none of the patients with available imaging in either group had radiographic findings consistent with gross loosening of the glenoid baseplate at two year followup despite the majority of cases (75%) using 6 or 8 mm of lateralization. One explanation for this may be the modularity baseplate design used in this study, which allowed for the placement of long central posts or screws. Biomechanical investigation, for instance, has shown that longer central posts improve fixation at time zero [13,14]. In our series, the most common lengths of the central screw or post were 25 and 20 mm, respectively, which may explain the low level of radiographic loosening. Due to the low level of loosening, it is difficult to make any conclusions regarding the influence of the difference in the β-angles in the two groups. Overall, our data suggest that both central screws and central posts are adequate for fixation when considering a lateralized design; however, these findings may not apply to the use of shorter screws or posts.

The total number of complications in our series was small, with a revision rate of only 1.4%. No significant differences were noted between the CP or CS groups, although as mentioned one patient with a central post did require revision for glenoid loosening. In our study, there were no reported acromial stress fractures despite lateralization as high as 8 mm. We attribute this to the inlay humeral component design, which is associated with a lower rate of stress fracture when compared to an onlay design [15].

There were several weaknesses of this study. This is a retrospective study and, thus, suffers from the consequences typical of retrospective reviews. Most significantly, over half of the study subjects did not have full radiographs available at two years for review. This is a consequence of the data coming from a registry database, in which radiographic imaging was not standardized across all site locations. In addition, while patient reported outcomes were standardized, various clinicians obtained range of motion measurements, which could have led to variability across sites. Second, the clinical and radiographic followup was short-term at 2 years. With increased followup, it is possible that further loosening could occur as could other complications. Ideally, we plan to follow these patients, and additional studies can be performed to establish any long-term clinical sequelae of glenoid lateralization with post or screw baseplate fixation. Third, the results are not generalizable to the use of bone grafts, which were excluded from the analysis. Lastly, several surgeons were included in this analysis, and we cannot control for surgical technique and other surgeon-related factors.

## 5. Conclusions

When using a lateralized implant with a 135 NSA, both central post and central screw baseplate fixation provided good clinical outcomes, survivorship, and improvements in ROM at 2 years. There did not appear to be a difference in loosening or revision rates between the two constructs.

## Figures and Tables

**Figure 1 jcm-11-03763-f001:**
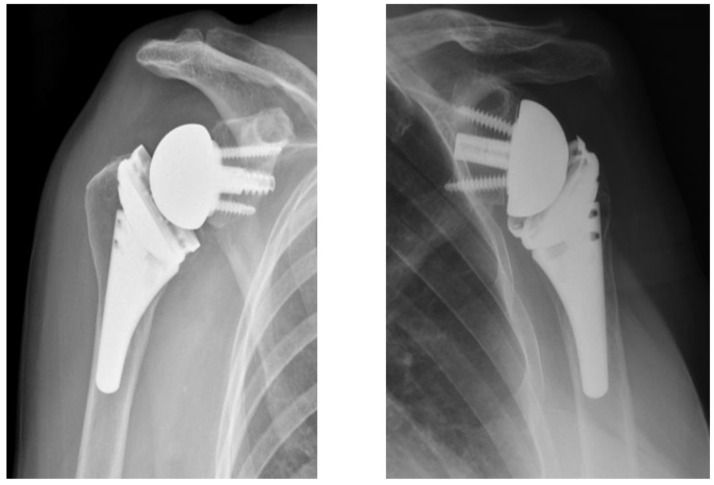
Radiographic demonstration of central screw and central post modular glenoid baseplates.

**Table 1 jcm-11-03763-t001:** Demographic Comparison of RSA Patients With Central Screw versus Post.

Patient Characteristics	Overall (n = 212)		Central Screw (n = 125)		Central Post (n = 87)		*p*
Demographics							
Age: years (mean, s.d.)	68.2	8.3	68.0	9.5	68.5	6.3	0.668
Sex: male (n, %)	116	54.7%	61	48.8%	55	63.2%	0.038
BMI: kg/m^2^ (mean, s.d.)	29.6	6.1	29.3	6.6	30.0	5.4	0.415
Dominant arm: yes (n, %)	129	60.8%	79	63.2%	50	57.5%	0.401
Tobacco use: yes (n, %)	15	7.1%	9	7.2%	6	6.9%	0.932
Surgical/Implant							
Glenosphere Diameter							
33 mm (n, %)	37	17.5%	28	22.4%	9	10.3%	0.023
36 mm (n, %)	56	26.4%	32	25.6%	24	27.6%	0.747
39 mm (n, %)	85	40.1%	47	37.6%	38	43.7%	0.374
42 mm (n, %)	34	16.0%	18	14.4%	16	18.4%	0.436
Glenoid Lateralization							
2 mm (n, %)	5	2.4%	4	3.2%	1	1.1%	0.333
4 mm (n, %)	47	22.2%	27	21.6%	20	23.0%	0.811
6 mm (n, %)	94	44.3%	59	47.2%	35	40.2%	0.315
8 mm (n, %)	66	31.1%	35	28.0%	31	35.6%	0.238
Other Surgical							
Central Screw/Post length, mm (mode, range)	25	15–35	25	15–35	20	15–30	<0.001
Peel/Tenotomy (n, %)	156	73.6%	78	62.4%	78	89.7%	<0.001
CT-based Preoperative Planning (n, %)	49	23.1%	26	20.8%	23	26.4%	0.338

**Table 2 jcm-11-03763-t002:** Comparison of Baseline PROs and ROM.

Patient Characteristics	Overall (n = 212)		Central Screw (n = 125)		Central Post (n = 87)		*p*
Baseline PROs	Mean	Std. Dev.	Mean	Std. Dev.	Mean	Std. Dev.	
ASES	41.8	17.8	41.7	18.8	42.0	16.3	0.904
VAS Pain	5.4	2.6	5.4	2.8	5.4	2.3	1.000
SANE	32.1	21.7	28.5	20.3	37.2	22.6	0.004
Constant-Murley	33.2	13.2	32.3	14.7	34.3	11.1	0.284
WOOS	38.0	19.0	37.7	20.1	38.5	17.2	0.763
VR-12 Mental	49.1	11.4	48.0	11.7	50.6	10.8	0.102
Baseline ROM							
Active FF (degrees)	93	36	91	38	97	32	0.230
Active ER at Side (degrees)	28	22	26	25	31	16	0.102
Active ER at 90 (degrees)	31	27	30	30	32	23	0.601
Active IR (spinal level)	L5	3	L5	3	L5	2	1.000
Active IR at 90 (degrees)	23	24	27	28	18	17	0.006

**Table 3 jcm-11-03763-t003:** Comparison of Two Year Clinical Outcomes.

Patient Characteristics	Overall (n = 212)		Central Screw (n = 125)		Central Post (n = 87)		*p*
2-Year PROs	Mean	Std. Dev.	Mean	Std. Dev.	Mean	Std. Dev.	
ASES	82.6	18.9	81.8	19.5	83.8	17.8	0.448
VAS Pain	1.2	2.1	1.3	2.2	1.0	2.0	0.312
SANE	73.9	25.6	71.7	27.5	77.2	22.3	0.124
Constant-Murley	67.9	12.1	67.1	12.5	68.6	11.7	0.379
WOOS	84.5	19.9	83.0	21.7	86.7	16.6	0.182
VR-12 Mental	53.2	9.1	53.0	9.2	53.6	8.9	0.636
2-Year ROM							
Active FF (degrees)	137	22	133	24	142	18	0.003
Active ER at Side (degrees)	46	15	46	18	45	12	0.651
Active ER at 90 (degrees)	66	23	63	25	69	20	0.064
Active IR (spinal level)	L4	3	L5	3	L3	3	<0.001
Active IR at 90 (degrees)	40	20	41	21	39	19	0.479

**Table 4 jcm-11-03763-t004:** Comparison of Change from Pre-op to Post-op.

Patient Characteristics	Overall (n = 212)		Central Screw (n = 125)		Central Post (n = 87)		*p*
2-Year PROs	Mean	Std. Dev.	Mean	Std. Dev.	Mean	Std. Dev.	
ASES	40.6	22.9	40.0	24.2	41.6	20.8	0.617
VAS Pain	-4.2	3.2	−4.0	3.4	−4.4	2.8	0.367
SANE	41.7	32.7	43.0	33.4	39.7	31.5	0.470
Constant-Murley	35.7	17.5	36.8	21.1	34.8	13.5	0.436
WOOS	46.3	25.0	45.2	26.8	47.8	22.0	0.456
VR-12 Mental	4.1	11.5	5.1	11.8	2.8	10.9	0.151
2-Year ROM							
Active FF (degrees)	48	33	50	39	46	27	0.409
Active ER at Side (degrees)	19	21	24	21	15	21	0.002
Active ER at 90 (degrees)	36	31	34	34	37	28	0.498
Active IR (spinal level)	1	4	0	4	2	3	<0.001
Active IR at 90 (degrees)	21	30	18	35	23	25	0.254

## Data Availability

The data is not publicly available.

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
