# Peer review of "A Comparison of Central Screw versus Post for Glenoid Baseplate Fixation in Reverse Shoulder Arthroplasty Using a Lateralized Glenoid Design"

_jcm, 2022, doi:10.3390/jcm11133763_

Round 1
Reviewer 1 Report
Page 3, line 92-93: it is stated that the "rate of revision at 2-year follow up was also calculated" - however, on page 2; line 59 it says that "revision RSA" was an exclusion factor - I think the authors need to clarify this, it is either an exclusion or not.
Page 4; line 131: it is stated that 3 patients were excluded due to revision, leaving 209 as a sample, however in both Table 2 and 3 the sample is still the original 212? Please clarify and rectify where necessary
Page 5; Table 2: firstly, may be an editing thing, but Table headings should be at the top of the table. Secondly, for the Active IR (spinal level), the columns for Mean just say L5 - I am assuming this is a typo - please rectify
Page 5; line 141: it is stated that no significant changes were seen except increase in active FF (142 vs 133; p = 0.003), however this does not correlate to what is in the table 3? I can't find these values? Table 3 says the active FF is 50 (CS) vs 46 (CP) with a p-value of 0.002. - please clarify or rectify
Page 5; lines 140-145 - there is a lot of repetition in these sentences, please consolidate.
Page 6; line 149: it is stated that only 99 individuals from the sample had 2 year follow up x-rays - however in the outline for the inclusion and exclusion criteria, again it says that that minimum of 2 year follow up x-rays are part of the inclusion. Perhaps adapt this to be more say that lack of 2 year follow up imaging was not considered an exclusion criteria? Saying this, this is considered a large LTFU - it is almost a 53% loss to follow up for this criteria evaluated and should be considered a very high risk in terms of bias due to missing outcome data - this should be stressed further to what is mentioned in the limitations already and cautioned in the discussion
General results - since this is a study comparing two "methods" a table that illustrates or demonstrates these comparisons should be included. Table 3 is labelled "comparison of change" however, this is the just the results for the 2 year follow up, there is no "comparison" in the table at all - the readers have to eyeball between table 2 and 3 to compare. Additionally, it would be better to be able to see if there was significant difference between the pre- and post groups - this statistic does not seem to have been run?
Page 6; line 152-154: it is stated that there is a difference in the B-angles between the CS and CP groups - however this is not addressed in the discussion. Since this has been linked to influence the implant loosening, perhaps the authors can elaborate on this aspect critically in their discussion.
Page6; line 168 - the first line of the discussion states that revision rates are low for both groups. However, since "revision" was considered an exclusion factor for this study, this statement cannot be the primary finding. If you excluded revision patients, then it is obvious revision rates would be low? This needs to be clarified and rectified if necessary.
Page 6; line 185: it is stated that in both the CS and CP groups that there was significant improvement - however there is no "statistical" data presented in the results to support this statement (see earlier comment for results)
Page 6; line 186: it is stated that their is significant difference in the 2 year follow up group in favor of CP - however there is no difference between active FF between CS and CP groups in table 3 (p = 0.409) - please rectify
Page 7; line 199-209: I think the authors should be careful about making these inferences regarding base plate loosening since more than 50% of the study cohort did not have x-ray imaging at the 2 year follow up.
Page 7; line 210-211: again, stating revision was only 1.4%, when one of your exclusion criteria was revision - does not correlate well
Page 7; line 222-224: is it currently known at what stage these baseplates tend to loosen? Perhaps, if there is literature in this regard to add a mention for this - if loosening is correlated with time after procedure, this could be an interesting aspect to consider.
Reviewer 2 Report
Thank you for the opportunity to review this study for Journal of Clinical Medicine.
In this study, the authors report the functional and radiological outcome of reverse shoulder arthroplasty using a lateralized glenoid construct with either a central screw or post. The results showed that there was no significant difference in short-term postoperative functional outcome and loosening or revision rate between groups using central screw and central post. This study is excellent in that it provides a detailed evaluation of range of motion and functional scores. This manuscript is written in very easy-to-understand English. The statistical analysis used in this study is also considered valid. However, as noted in the Discussion section, the most disappointing aspect of this study is that more than half of the cohort did not have plain radiography to assess radiological outcomes at 2 years postoperatively. Since this study was designed to evaluate the stability of glenoid baseplate fixation (central post or central screw), it is necessary to evaluate whether or not radiographic changes such as loosening occur in the long term. It is a serious limitation that the majority of patients lack an evaluation of whether signs suggestive of loosening have already appeared, even if short-term postoperative function is good. The authors mention that they are planning to evaluate long-term outcome in limitation, but evaluation of the radiological outcome using plain radiography is still necessary in this case.
As a minor comment, the range of motion of active IR in Table 3 is missing the notation of spinal levels such as Th and L.
Round 2
Reviewer 1 Report
Thank you for addressing all the comments
Reviewer 2 Report
Other than the points I raised during the last peer review, I see no problems.